# Antibiotic Treatment Prior to Injury Improves Post-Traumatic Osteoarthritis Outcomes in Mice

**DOI:** 10.3390/ijms21176424

**Published:** 2020-09-03

**Authors:** Melanie E. Mendez, Deepa K. Murugesh, Aimy Sebastian, Nicholas R. Hum, Summer A. McCloy, Edward A. Kuhn, Blaine A. Christiansen, Gabriela G. Loots

**Affiliations:** 1Lawrence Livermore National Laboratories, Physical and Life Sciences Directorate, Livermore, CA 94550, USA; mendez20@llnl.gov (M.E.M.); murugesh2@llnl.gov (D.K.M.); sebastian4@llnl.gov (A.S.); hum3@llnl.gov (N.R.H.); mccloy2@llnl.gov (S.A.M.); kuhn7@llnl.gov (E.A.K.); 2UC Merced, School of Natural Sciences, Merced, CA 95343, USA; 3UC Davis Medical Center, Department of Orthopedic Surgery, Sacramento, CA 95817, USA; bchristiansen@ucdavis.edu

**Keywords:** osteoarthritis, PTOA, gene expression, RNA-seq, cartilage degeneration, tibial compression, gut microbiome, antibiotics, LPS

## Abstract

Osteoarthritis (OA) is a painful and debilitating disease characterized by the chronic and progressive degradation of articular cartilage. Post-traumatic OA (PTOA) is a secondary form of OA that develops in ~50% of cases of severe articular injury. Inflammation and re-occurring injury have been implicated as contributing to the progression of PTOA after the initial injury. However, there is very little known about external factors prior to injury that could affect the risk of PTOA development. To examine how the gut microbiome affects PTOA development we used a chronic antibiotic treatment regimen starting at weaning for six weeks prior to ACL rupture, in mice. A six-weeks post-injury histological examination showed more robust cartilage staining on the antibiotic (AB)-treated mice than the untreated controls (VEH), suggesting slower disease progression in AB cohorts. Injured joints also showed an increase in the presence of anti-inflammatory M2 macrophages in the AB group. Molecularly, the phenotype correlated with a significantly lower expression of inflammatory genes *Tlr5, Ccl8, Cxcl13,* and *Foxo6* in the injured joints of AB-treated animals. Our results indicate that a reduced state of inflammation at the time of injury and a lower expression of Wnt signaling modulatory protein, *Rspo1,* caused by AB treatment can slow down or improve PTOA outcomes.

## 1. Introduction

The individual microbial cells that constitute the human gut microbiome outnumber our cells by a factor of 10 [1]. In utero, most fetuses are free of microorganisms [2]; the first exposure babies have to microbes is during birth as they move through the birth canal, hence babies born in natural birth are inoculated with microorganisms by their mothers. The gut microbiome initiates with breast feeding and builds complexity as the baby’s diet evolves from milk to other types of foods [3]. It reaches dynamic stability by the age of 3, and while a person’s gut biome is relatively stable, there are many genetic and environmental factors that influence its composition and dynamic change in each person [4]. The microorganisms living in our gut that do not cause harm, and may even have a beneficial contribution to human health, are called commensals. The gut microbiome composition can be disrupted by dietary changes, antibiotic treatment or pathogenic infections, and in reciprocal interactions changes to the composition and abundance of commensals could affect the entire system by producing unwarranted gastrointestinal and immune diseases [5,6,7]. 

Published literature suggests that the gut microbiome has an indirect effect on bone through changes to the immune system and inflammatory cytokines [8,9]. Commensals aid in immune-regulation by releasing microbial associated molecular patterns (MAMPs) such as lipopolysaccharides (LPSs) that bind and activate toll-like receptors (*Tlr*); LPSs have been shown to bind to *Tlr4* [9,10,11,12,13,14]. This activation causes an inflammatory cascade that releases inflammatory cytokines and interferons which act as transcription factors to induce naïve immune cells to mature [15,16,17]. Studies have suggested that gut biome dysbiosis can promote aggressive bone destruction mediated by osteoclasts due to an increase in tumor necrosis factor alpha (TNF-α) [18,19,20]. Furthermore, we have previously shown that elevated levels of LPSs negatively impact bone by promoting bone loss and accelerate post-traumatic osteoarthritis (PTOA) development. We also showed that LPS administration prior to injury elevates *Tlr5/7/8* transcription in the joint [10,21]. TNF-α promotes osteoclastogenesis by increasing RANK-L expression in bone marrow cells and therefore elevating the number of osteoclast precursor cells [20,22,23,24]. 

In the context of PTOA, it has been shown that when the gut microbiome of obese mice is modified by supplementing oligofructose, OA phenotypes diminish, which correlates with a reduction in the levels of inflammation in the colon and cytokine levels circulating in the serum and present in the knee [25]. Cyclic compressive loading in mice on a high-fat diet have promoted more severe PTOA phenotypes than mice on a normal diet, while *Tlr5^-/-^* mice treated with ampicillin and neomycin have shown improvement in the cartilage phenotype post injury [26]. Germ-free mice have also been shown to have a better OA outcome after destabilization of the medial meniscus, and modifications to the gut microbiome have improved PTOA phenotype in obese mice [25,27,28]. Tibial compression induced injury in 20-week-old germ-free mice has shown an increase in bone volume [29]. Therefore, a precedent exists in support of the gut microbiome composition as a potential risk factor for the development of PTOA, but additional studies are required to elucidate potential mechanisms that contribute to the unwarranted PTOA phenotypes.

Most PTOA-related studies to date have examined factors likely to exacerbate or accelerate the development of osteoarthritis post injury, if administered at the time of injury or shortly thereafter [30]. Since bone and cartilage sometimes exhibit an inverse relationship to insult, such that what is anabolic for bone is catabolic for cartilage and vice versa [31], we speculated that gut biome modifications would slow down or improve PTOA outcomes. Therefore, this study aimed to examine how partial elimination of the gut microbiome through antibiotic treatment prior to injury would influence PTOA outcomes. Studying the effects of medication administered before an injury is of high biomedical importance because, clinically, most concerns are centered on side-effects due to co-administration. Currently, however, standards of medical care do not consider gut biome status, nor is gut dysbiosis a recoded clinical parameter. Research that can show the prognostic and diagnostic value of gut biome status could potentially lead to new standards of care. In addition, antibiotics are widely prescribed to teens and young adults who may be active in sports and therefor more susceptible to joint injuries. According to the CDC, in 2016, 64.9 million oral antibiotic prescriptions were issued to people under the age of 20, the equivalent of 790 per 1000 people; therefore, gut dysbiosis may be more common than expected in young athletes suffering an articular injury [32]. As the population of the USA ages there will be an increase in PTOA cases; studying how antibiotics modify PTOA phenotypes will be helpful to finding preventative treatments in the future for both young and old patients. 

## 2. Results

### 2.1. Antibiotic Treatment Prior to Injury Delays Cartilage Resorption in Injured Joints

Using an established, noninvasive, tibial compression PTOA mouse model [33,34,35,36,37], we examined whether a six-week course of antibiotics (ampicillin (1.0 g/L)/neomycin (0.5 g/L)) [26] would impact OA outcomes, post injury. C57Bl/6J mice were examined histologically at six weeks post injury. Examination of the uninjured, contralateral femoral heads revealed a more intense Safranin-O staining throughout the articular cartilage of the antibiotic (AB)-treated group compared to untreated controls (VEH), but both AB and VEH joints displayed normal morphology (Figure 1A,C). A slightly less mineralized area, characterized by large pockets of bone marrow, was observed in the femoral condyle of AB-treated, injured joints, relative to the VEH injured group (Figure 1B,D). Consistent with the AB uninjured control, the femoral condyle of the injured AB group (Figure 1d; arrow, asterisk) appeared to also have an increase in Safranin-O staining intensity as well as a thicker articular cartilage layer than the injured VEH, suggesting higher levels of proteoglycans and reduced chondrocyte apoptosis in the injured joints of AB-treated animals (Figure 1b; arrow, asterisk). The meniscus (Figure 1bb,dd; arrow, asterisk) of injured AB joints showed a thicker hyperplastic morphology with enhanced cellular infiltration. The meniscus in the injured VEH group also showed cellular infiltration but at a significantly lower level than the AB injured group. Examination of the sagittal views of the joints by a modified Osteoarthritis Research Society International (OARSI) grading scale determined that AB-treated injured joints had a significantly lower cartilage score than VEH-treated injured joints with a *p*-value of 0.038 (Figure 1E). These results imply that modifying the gut microbiome through the administration of an ampicillin/neomycin antibiotic cocktail prior to injury was sufficient to improve the cartilage phenotype subsequent to trauma, and reduce PTOA outcomes. 

### 2.2. Antibiotic Treatment Has a Negative Effect on Bone, Post Injury

The bone phenotypes of AB- and VEH-treated mice were characterized by micro-computed tomography (µCT) to quantify subchondral trabecular bone mass and osteophyte volume at six weeks post injury. Consistent with prior published results, VEH injured joints had significantly less subchondral bone volume (BV/TV) by ~17.28% and ~11.31% when compared to contralateral and uninjured controls, respectively (Figure 2A). Antibiotic-treated injured joints lost 13.67% when compared to contralateral AB treated and 17.42% when compared to uninjured AB-treated controls. The subchondral bone volume (BV/TV) fraction of AB group had ~0.2%, ~7.1%, and ~10.9% lower BV/TV than the VEH group when comparing the uninjured, injured, and contralateral groups; the contralateral group was the only one that was statistically significant, suggesting that AB treatment does not elevate BV/TV in the uninjured leg (Figure 2A). Trabecular number (Tb.N) of the AB group had ~0.2% and ~0.4% higher Tb.N on the uninjured and injured groups, respectively, compared to the VEH; the contralateral had ~1.1% lower Tb.N on the VEH. Tb.N was not statistically significant (Figure 2B). Trabecular thickness (Tb.Th) showed the VEH group had ~2.9%, ~7.6%, and ~10.6 higher Tb.Th than uninjured, injured, and contralateral AB cohorts; injured and contralateral were significant (Figure 2C). Trabecular spacing (Tb.Sp) AB showed ~1%, ~0.4%, and ~2.2% larger Tb.Sp than the VEH uninjured, injured, and contralateral groups, respectively; none were statistically significant (Figure 2D). The VEH group had ~39.7% significantly higher osteophyte volume (Op.V) compared to the AB cohort (Figure 2E). Visual representations of Op.V showed a larger amount of osteophytes in VEH joints, consistent with the quantification data (Figure 2F).

### 2.3. LPS Treatment Compared to AB Treatment

Lipopolysaccharides (LPSs) administered five days prior to the joint injury did not elicit any significant changes in the bone phenotype when compared to the AB-treated uninjured bones. However, as previously reported [10], LPS administration alone was sufficient to modulate the cartilage phenotype on both the contralateral and injured joints towards a more severe phenotype (Figure 3E,F). LPS injured joints showed an enhanced thinning of the femoral cartilage that was distinguishable from the cartilage of the VEH and AB injured groups (Figure 3b,d,f). The damage to the cartilage in the LPS injured joints corresponded with a significantly higher OARSI score than the VEH and AB injured joints (Figure 3I). There was an increase in cellular infiltration on the AB- and LPS-treated joints compared to the VEH (Figure 3bb,dd,ff). Combination of the AB treatment and LPS challenge (AB+LPS) significantly improved the LPS-mediated phenotype, reverting the contralaterals back to the OARSI scores recorded for the VEH injured and uninjured groups (Figure 3A,B,G,H). These results indicate that the effects of LPSs are blunted by the AB treatment, where statistical analysis does not distinguish between VEH and AB+LPS in either the uninjured or injured groups (Figure 3I). 

#### Macrophages Associated with Healing Are Increased in Antibiotic-Treated Joints

Tissue resident macrophages are essential in providing innate immune defenses and regulating tissue and organ homeostasis [39]. Macrophages and other inflammatory cells are recruited to injury sites where they also play key roles in tissue remodeling and repair [40,41]. In the joint there is a population of inactive macrophages residing in the synovium [42]. These cells are activated under certain conditions such as injury or inflammation. Macrophages can be found using the marker F4/80, which is a marker for cells of mononuclear phagocyte lineage in mice [43]. Macrophages can be both pro-inflammatory (M1) and anti-inflammatory (M2), and the co-action of these subtypes can repair damaged tissue through specific cytokine secretion. Identification of both macrophage populations was done using F4/80 and iNOS as an M1 marker while using CD206 as a M2 marker. In the joint, however, a change in the M1/M2 ratio may be critical in PTOA progression and development. Histological examination of the injured joints indicated a hyperplastic synovium that appeared to have significant cellular infiltration on LPS- and AB-treated injured joints (Figure 3dd,ff; asterisk), and this morphology was similar to that of LPS-treated injured joints previously described [10]. To determine whether AB treatment prior to ACL rupture altered the composition of M1 and M2 cells in the injured joint, we used M1/M2 specific antibody markers to distinguish these subtypes by immunohistochemistry (Figure 4). On the meniscus, we observed a slightly higher staining of anti iNOS antibody indicative of some M1 macrophage infiltration in the AB injured joints compared to the VEH injured joints (Figure 4A,B). The M1 macrophage infiltration presence in AB injured joints was less than that of LPS injured joints (Figure 4B,C). In contrast, we observed a higher level of anti CD206 antibody staining in AB injured joints when compared to VEH and LPS injured, indicative of higher levels of M2 macrophages present in the injured joints of AB-treated animals (Figure 4D–F). These results suggest that while there is an increase in macrophages for both LPS and AB treatments, the LPS-treated joints have an increase in pro-inflammatory macrophages while AB-treated joints have an increase in anti-inflammatory macrophages that may be helping mitigate the PTOA phenotype.

### 2.4. Gene Expression Changes Associated with Chronic Antibiotic Treatment 

To determine antibiotic treatment-related molecular changes in knee joints we compared the transcriptome of 10-week-old mice that had been treated with AB in their drinking water to VEH controls. We found 620 significantly upregulated genes in uninjured AB compared to the uninjured VEH group; the majority of genes accounting for these transcriptional changes were associated with immune responses. Some of the groups we found were of genes associated with collagen (*C1qb* [44], and *Fcna* [45]), regulators of adaptive immunity (*Btla* [46], *Lax1* [47], *Tnfrsf13c* [48], *Lat* [49], and *Cd40* [50]), and genes associated with the major histocompatibility complex type II (*Tnfrsf14* [51], *Cd86* [52], and *Cd8b1* [53]). We found 737 significantly downregulated genes in the AB uninjured group compared to the VEH uninjured group. These genes included regulators of skeletal development (*Alpl* [37], *Col6a1* [54], *Col6a2* [55], *Sox9* [37], *Sox11* [56], and *Wnt9a* [57]), Hippo signaling (*Tead1*, *Tead4* [58], and *Yap1* [59]), muscle contraction (*Myh8*, *Myh2*, and *Myom2*), and inflammatory response (*Tlr5* [60], *Ccl8* [61], *Cxcl13* [37], *C5ar1* [62], *Cxcr1* [63], and *Foxo6* [64]). Inflammatory gene comparison between AB injured and uninjured groups compared to the corresponding VEH groups are shown in Figure 5A and Table 1.

We also found 185 genes significantly upregulated when comparing the AB injured to the VEH injured group. Among these we found negative regulators of cytokine production (*Arg1* and *Sars*) and small molecule catabolic processes (*Bad* and *Galk1*). We found 284 genes significantly downregulated when comparing the AB injured group to the VEH injured group. Among these we found regulators of the Wnt signaling pathway (*Sost* [35,65], *Fzd3*), Notch signaling pathway (Notch4), collagen-associated genes (*Chad*, *Col16a1*, and *Lep*), and transcription factors known to be involved in cellular differentiation processes (*Sox18*, *Foxd2*, and *Sox10*). Genes associated with bone and cartilage formation are shown in Figure 5B and Table 2.

There were 78 genes upregulated in both AB groups when compared to their corresponding VEH control groups, including *C1qb* [44] and *Cd3d* [66]. There were 151 genes downregulated in both AB groups compared to the VEH group. These included skeletal muscle contraction (*Myh1*, *Myh3* [67], *Myh14* [68], and *Myom3* [69]), and extracellular matrix proteins found in bone and cartilage (*Col1a1*, *Col2a1*, *Col5a1*, *Col6a3* [37], and *Col7a1* [70]). *Tbx5* and *Klhl40* were upregulated in the AB injured group and downregulated in the AB uninjured group when compared to their corresponding VEH groups. *Cd209a* and *Rspo1* were downregulated in the AB injured group and upregulated in the AB uninjured group when compared to their corresponding VEH groups. Comparing the AB uninjured and AB injured groups showed that bone formation genes like *Cthrc1* were upregulated in the AB injured group. *Mmp3* [37], *Lox*, *Loxl3*, *Mmp10*, and *Mmp19*, were downregulated in the AB uninjured group when compared to the AB injured group.

#### Comparison of Gene Expression Changes Between Chronic Antibiotic Treatment and LPS Treatment 

In order to compare treatment-related molecular changes in the joint we compared the expression of AB and LPS to VEH of the same injury. We found 32 genes that were upregulated in both the AB and LPS groups compared to the uninjured VEH group. Among these we found genes related to the inflammatory and immune system (*Cx3cr1*, *Cd3d*, *Ccl4*, and *Ccl6*). We found *Rspo1* [71] to be upregulated in both the AB and LPS groups compared to the VEH groups, and downregulated in the AB injured group when compared to the injured VEH group. There were three genes (*Fgfbp1*, *Tnnc1*, and *Rab20*) found to be upregulated in the injured AB and LPS groups compared to the injured VEH. We found 18 genes downregulated in the AB and LPS uninjured groups when compared to the corresponding VEH groups. Genes related to the immune system were among these (*Bcl3*, *Prtn3*, *Tnfsf9*, and *Ifitm1*). We found seven genes to be upregulated in the uninjured LPS group while downregulated in the uninjured AB group when compared to the corresponding VEH groups. These genes are *Tlr5*, *C5ar1*, *Aqp4*, *Ryr3*, *Mdga1*, *Foxo6*, and *Kcng4*. We found two genes (*Cd209a* and *Lep*) downregulated in both injured the AB and LPS groups compared to the injured vehicle. We found marker *Cd209a*, which is present in macrophages and dendritic cells, to be upregulated in the uninjured AB and LPS groups and downregulated in the injured AB and LPS groups when compared to the corresponding VEH groups. Gene expression between the AB and LPS groups compared to the VEH groups is found in Table 3.

## 3. Discussion

The role of chronic antibiotic treatment prior to injury on the development of PTOA has not been previously investigated. Previous studies associating gut biome changes with skeletal phenotypes have focused primarily on bone, and have shown that the gut microbiome can influence osteoclastogenesis and affect bone volume [23]. Previous studies have also described the changes in the gut microbiome and bone using the same antibiotic cocktail [26,72]; however, our study is the first to examine how depletion of the gut biome with an ampicillin/neomycin antibiotics cocktail affects the development of osteoarthritis following a traumatic joint injury. Although morphologically our data indicate increased cellular infiltration in the synovium of injured joints of antibiotic-treated cohorts, reminiscent of synovitis induced by LPS administration [10], AB treatment did not have a significant effect on the BV/TV of the injured joint, resulting in no significant changes in BV/TV between the injured VEH and injured AB groups. The only significant bone phenotype was observed when contralateral were compared, where VEH-treated mice had an increase in BV/TV relative to uninjured control AB-treated mice had a slight, but significant decrease in BV/TV. However, AB treatment improved the cartilage phenotype as reflected by a significantly lower OARSI score in these mice. There are several possible explanations for this outcome. The cellularity was examined using Ly6G and Ly6C markers, which highlighted an increase in the presence of monocytes, neutrophils, and granulocytes. In order to increase specificity, we stained with M1 and M2 markers, which showed an increase in the frequency of pro-inflammatory macrophages in the LPS-treated joints, while more anti-inflammatory macrophages were present in the AB-treated group. This correlation supports the conclusion that different macrophage subtypes could have influenced the exacerbation of PTOA in the LPS-treated cohorts, while deaccelerating PTOA progression in AB-treated cohorts. 

It is also possible that chronic antibiotic treatment in juvenile mice (starting at four weeks of age) is anabolic to the articular cartilage, and during the six weeks of AB treatment prior to the injury the cartilage of these animals produced significantly more extracellular matrix. Furthermore, the cartilage of AB-treated animals may also display slightly different biomechanical properties than VEH controls. This theory is in part supported by the observation that uninjured joints of AB-treated animals stained more intensely with Safranin-O than the VEH uninjured joints (Figure 1A,C). If the cartilage of AB-treated animals acquired different mechanical properties, including increased stiffness or increased elastic modulus, cartilage degradation in these mice may have proceeded at a slower pace than the VEH, accounting for the milder phenotype in these injured joints. Additionally, we see a downregulation of gene *Rspo1* in injured AB joints when compared to injured VEH. This could be the reasoning for the decrease in OA progression, as *Rspo1* has previously been shown to have a role during OA progression [71]. Future studies examining the biomechanical properties of cartilage at different ages and in different treatment groups will have to be conducted to confirm whether significant differences in these properties exist. 

Alternatively, the observed PTOA outcomes may be driven primarily by molecular changes. We observed 113 genes, including inflammatory genes *Bmper*, *Ccl2*, *Ccl7*, *Ccl8*, *Cxcl5*, *IL6*, *IL11*, *IL33*, and *Cxcl10,* that overlapped between the injured VEH, LPS, and AB groups and the uninjured groups of their respective treatments with significantly elevated expression in the injured joints; however, the blunted effect was present only in the AB injured group, suggesting that these molecules have a less potent effect in when mice are treated with AB prior to injury. Gene expression data indicate that the inflammatory genes *Ptgs2, Reg3g,* and *Serpine1* were downregulated in uninjured AB-treated mice, while *Tbx21* was found to be downregulated in injured AB joints. Two inflammatory genes found to be downregulated in both the injured and uninjured AB groups as compared to the corresponding VEH were *Tafa3* and *Cntfr*, which are both associated with the immune and nervous system. These genes have not been studied in the context of PTOA, and their influence on injury outcome would be interesting to study. When examining molecular changes in the immune system in prior reports, we have shown that elevated and persistent immune activation accelerates osteoarthritis phenotypes post injury [10]. Furthermore, we have shown that systemic LPS administration five days prior to injury negatively impacts PTOA, resulting in a more severe phonotype [10]. We have also shown that LPS-treated mice display highly elevated levels of toll-like receptors 5, 7, and 8 (*Tlr5*, *Tlr7*, *Tlr8*), and we hypothesized that the enhanced PTOA phenotype in LPS-treated mice may be due in part to increased signaling through these receptors. One complementary transcriptional result derived from the RNA-seq data examined herein is the discovery that *Tlr5* is significantly downregulated in AB-treated uninjured joints. Kim et al. showed that *Tlr5* in rheumatoid arthritis promotes monocyte presence and osteoclast formation due to the cross regulation of the *Tlr5* and TNF-α pathways [73]. If *Tlr5* similarly modulates the expression of inflammatory genes that are directly involved in cartilage degradation post injury, the observed transcriptional suppression of *Tlr5* in uninjured joints may promote a molecular resistance to inflammatory cytokines. Future studies will have to evaluate whether *Tlr5* receptor antagonists can prevent or slow down the development of OA post ACL rupture.

The current literature presents conflicting evidence on the effects of antibiotic treatment and the gut microbiome on bone, showing that modifications may have no significant changes in BV/TV after injury when compared to untreated control mice, but have a lowering effect when compared to uninjured treated mice [26]. In germ-free mice, AB treatment can increase the BV/TV after injury compared to injured controls, but these mice also have a lower BV/TV when compared to uninjured germ-free mice [29]. Our results showed no changes in BV/TV on injured AB-treated mice when compared to the injured VEH, but did show a decrease in in BV/TV when compared to the AB contralateral. These results are similar to the decrease in BV/TV observed after tibial compression (TC) injury in strains that are resistant to PTOA, like *MRL/MpJ* [36,74]. Our AB group showed an improvement in cartilage, and though these mice are not OA-resistant to the extent of *MRL/MpJ* mice, we observed similarities in the expression of T-cell markers like *Cd3d* and *Cd8b1* [36]. Although inflammation is resolved quicker in *MRL/MpJ* mice, bone resorption still occurs, similar to the C57Bl6 strain, due to the presence of pro-inflammatory cytokines and matrix metalloproteinases (MMPs) that promote osteoclastogenesis and extracellular matrix degradation [75,76,77]. The similarity of T-cell markers could potentially enhance healing and accelerate inflammation resolution in AB-treated mice, which could diminish the PTOA phenotype. 

Our study uniquely examines the impact of long-term antibiotic treatment on OA outcomes subsequent to joint trauma. Prior to this study, we did not know if there were any PTOA changes caused by the administration of antibiotics. While we found that this particular AB regime had a beneficial effect on the health of injured and uninjured joints, it still remains to be elucidated whether short term AB treatment can be prophylactic; most importantly, questions remain about whether AB treatment post injury would have the same beneficial effect. This study highlights the importance of how the body works as a system and how systemic and local factors present prior to injury can significantly impact how our body heals and responds to trauma. This study highlights the importance of the gut biome in modulating PTOA, specifically by affecting toll-like receptors transcriptional levels that may in turn influence PTOA outcome after injury.

## 4. Materials and Methods 

### 4.1. Tibial Compression Overload

*C57Bl/6J* mice were purchased from the Jackson Laboratory (Bar Harbor, ME, USA; Stock No: 000664) at four weeks of age and randomized into experimental groups (n >= 4). The antibiotic (AB) -treated group received treatment (ampicillin [78,79] (1.0g/L; Sigma; A9518-25G; St. Louis, MO, USA); neomycin [80,81,82] (0.5 g/L; Sigma; N1876-25G; St. Louis, MO, USA) in drinking water starting at weaning (four weeks of age) for six weeks; the untreated group (VEH) was provided with regular drinking water. Five days prior to injury at 10 weeks of age, cohorts of mice were separated into three groups (VEH, AB, and lipopolysaccharide (LPS)). The LPS group received an intraperitoneal (IP) injection of LPS (1 mg/kg; Sigma; L6529-1MG; St. Louis, MO, USA), while the VEH and AB groups received an IP injection of saline of an equivalent volume. On the day of injury, all groups were subjected to non-invasive ACL rupture using a single dynamic tibial compressive overload using an electromagnetic material testing system (ElectroForce 3200, TA Instruments, New Castle, DE, USA) as previously described [35,74,83]. Cohorts were placed under anesthesia using isoflourane prior to injury [84]. ACL injury was performed by placing the mouse in the system and applying a compressive load at 1 mm/s until ACL rupture (typically 12N-14N); the uninjured group was placed in the system and received a non-injury compressive force (2N-3N). After injury, mice cohorts were given saline (0.05 mL) and buprenorphine (0.05 mg/kg) and returned to normal cage activity as previously described [34,35,36,74]. All animal experiments were approved by the Lawrence Livermore National Laboratory and University of California, Davis Institutional Animal Care and Use Committee (approved on 14/07/2016), and conformed to the Guide for the care and use of laboratory animals under protocol 250.

### 4.2. Micro-Computed Tomography (µCT) 

Injured joints, contralateral joints from injured mice, and bilateral joints from uninjured mice were collected six weeks post injury for all groups. Samples were dissected and fixed for 72 h at 4 °C using 10% neutral buffer formalin; samples were stored in 70% ethanol at 4 °C until scanned. Whole knees were scanned using a SCANO µCT 35 (Bassersdorf, Switzerland) according to the rodent bone structure analysis guidelines (X-ray tube potential = 55 kVp, intensity = 114 mA, 10 µm isotropic nominal voxel size, integration time = 900 ms) [34]. Trabecular bone in the distal femoral epiphysis was analyzed by manually drawing contours on 2D transverse slides to designate the region of trabecular bone enclosed by the growth plate and subchondral cortical bone plate. We quantified trabecular bone volume fraction (BV/TV), trabecular thickness (Tb.Th), trabecular number (Tb.N), and trabecular separation (Tb.Sp) [38]. Mineralized osteophyte volume in injured and contralateral joints was also quantified by drawing contours around all heterotopic mineralized tissue attached to the distal femur and proximal tibia as well as the whole fabellae, menisci, and patella. Total mineralized osteophyte volume was then determined as the volumetric difference in mineralized tissue between injured and uninjured joints. Statistical analysis was performed using two-way ANOVA and a Student’s *t*-test with a two-tailed distribution, with two-sample equal variance (homoscedastic test). For all tests, *p* < 0.05 was considered statistically significant. 

### 4.3. Histological Assessment of Articular Cartilage and Joint Degeneration

VEH-, AB-, and LPS-treated injured, contralateral, and uninjured joints were dissected six weeks post injury, then fixed, dehydrated, paraffin-embedded, and sectioned as previously described [74]. The cartilage was visualized in sagittal 6-µm paraffin serial using Safranin-O (0.1%, Sigma; S8884; St. Louis, MO, USA) and Fast Green (0.05%, Sigma; F7252; St. Louis, MO, USA) as previously described [35]. OA severity was evaluated using a modified Osteoarthritis Research Society International (OARSI) scoring scale as previously described [85]. Cartilage scoring began ~0.4 mm out from the start of synovium to the articular cartilage. Blinded slides were evaluated by seven scientists (six with and one without expertise in OA) utilizing modified (sagittal) OARSI scoring parameters due to the severe phenotype caused by TC loading-destabilization that promotes mechanical-induced tibial degeneration on injured joints [74,85]. Modified scores: (0) for intact cartilage staining with strong red staining on the femoral condyle and tibia; (1) minor fibrillation without cartilage loss; (2) clefts below the superficial zone; (3) cartilage thinning on the femoral condyle and tibia; (4) lack of staining on the femoral condyle and tibia; (5) staining present on 90% of the entire femoral condyle with tibial degeneration; (6) staining present on over 80% of the femoral condyle with tibial degeneration; (7) staining present on 75% of the femoral condyle with tibial degeneration; (8) staining present on over 50% of the femora condyle with tibial degeneration; (9) staining present in 25% of the femoral condyle with tibial degeneration; (10) staining present in less than 10% of the femoral condyle with tibial degeneration.

### 4.4. Immunofluorescent Staining

Six-micrometer sections from injured samples from both treatment groups of *C57Bl/6J* were used. Unitrieve was used as an antigen retrieval method for 30 min at 65 °C [86]. Primary antibodies: Anti-F4/80 (Abcam, ab16911(1:50)), Anti-CD206 (Abcam, ab64693(1:500)), and Anti-iNOS (Abcam, ab15323(1:100)) were used and incubated overnight at room temperature in a dark humid chamber. Negative control slides were incubated with secondary antibody only. Stained slides were mounted with Prolong Gold with DAPI (Molecular Probes, Eugene, OR, USA). Slides were imaged using a Leica DM5000 microscope. ImagePro Plus V7.0 Software and a QIClick CCD camera (QImaging, Surrey, BC, Canada) were used for imaging and photo editing.

### 4.5. RNA Sequencing and Data Analysis

*C57Bl/6J* injured and uninjured joints from VEH-, AB-, and LPS-treated male and female mice were collected 24 h after injury (n >= 4). Joints were dissected and cut at the edges of the joint region with small traces of soft tissue to preserve the articular integrity. RNA was isolated and sequenced as previously described [36]. RNA-seq data quality was checked using FastQC software (version 0.11.5). Sequence reads were aligned to the mouse reference genome (mm10) using STAR (version 2.6). After read mapping, ‘featureCounts’ from Rsubread package (version 1.30.5) was used to perform read summarization to generate gene-wise read counts. Differentially expressed genes (DEGs) were identified using edgeR (version 3.22.3). Genes with a false discovery rate (FDR) corrected *p*-value less than 0.05 and fold change greater than 1.5 were considered as DEGs. Heatmaps were generated using heatmap.2 function in R package ‘gplots’.

## 5. Conclusions

Chronic antibiotic administration known to deplete some gram negative bacteria, primarily Bacteroidetes, while expanding the Proteobacteria phylum provided protective benefit from the development of post-traumatic osteoarthritis after joint injury.


## Figures and Tables

**Figure 1 ijms-21-06424-f001:**
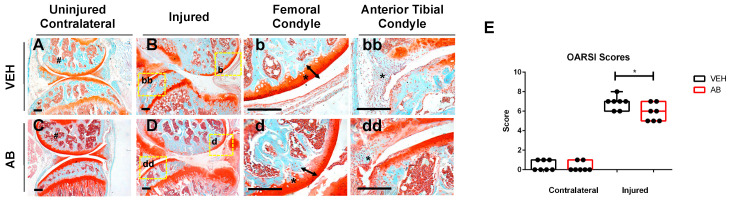
Characterization of post traumatic osteoarthritis (PTOA)-associated structural changes of antibiotic-treated animals in the knee. (**A**) Histological evaluation of vehicle (VEH) contralateral, (**B**–**bb**) VEH injured, (**C**) Antibiotic treated (AB) contralateral, and (**D**–**dd**) AB injured joints conducted at six weeks post injury using Safranin-O and Fast Green staining (scale bars indicate 200 mm). High magnification images corresponding to yellow boxes (**B**,**D**) are provided b, bb, d, dd. (**E**) PTOA severity was quantified using a modified OARSI scoring system (* *p*-value < 0.05).

**Figure 2 ijms-21-06424-f002:**
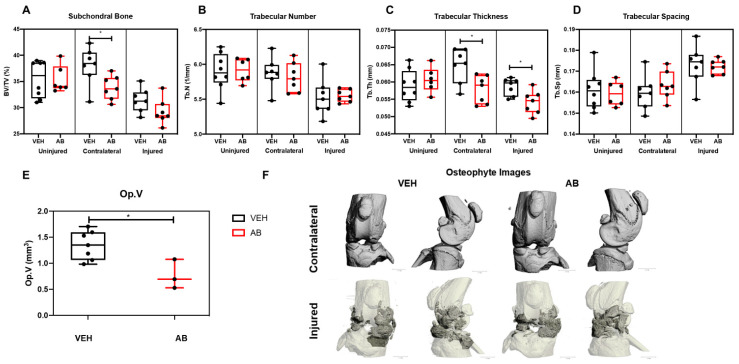
Bone phenotype of antibiotic-treated injured mice. (**A**) Subchondral trabecular bone volume fraction (BV/TV) of the distal femoral epiphysis. (**B**) Trabecular number was measured using the average number of trabeculae per unit length. (**C**) Trabecular thickness was measured using the mean thickness of trabeculae assessed using direct 3D methods. (**D**) Trabecular spacing was measured using the mean distance between trabeculae, assessed using direct 3D methods [38]. (**E**) Osteophyte volume at six weeks post injury. (**F**) Osteophyte imaging using µCT. (* *p* < 0.05).

**Figure 3 ijms-21-06424-f003:**
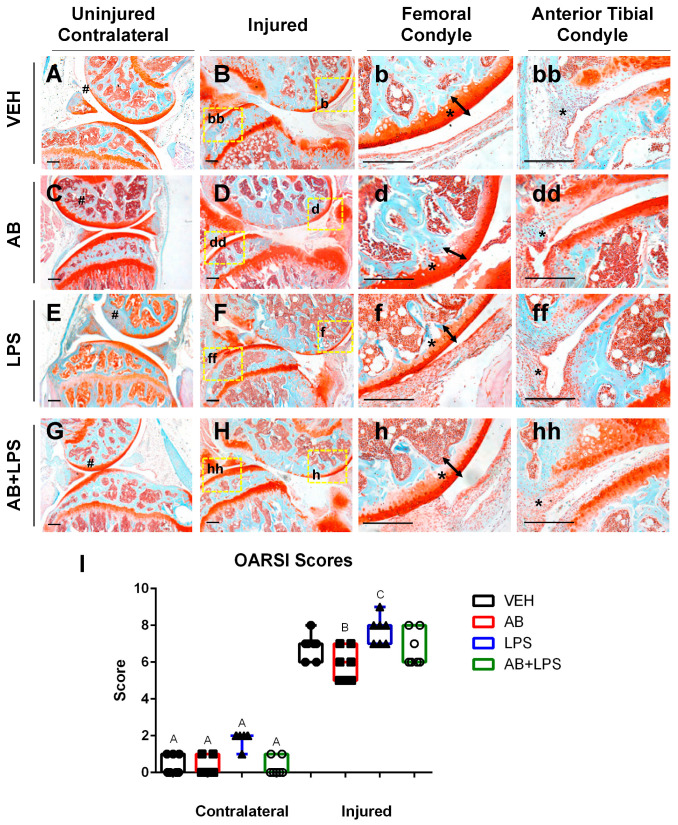
Characterization of PTOA-associated structural changes in the injured VEH, AB, LPS, and AB+LPS groups. (A–hh) Histological evaluation of uninjured and injured joints at six weeks post injury using Safranin-O and Fast Green staining. Black scale bars indicate 200 µm. Numeral sign indicates the femoral condyle (**A**,**C**,**E**,**G**). High magnification images corresponding to yellow boxes (**B**,**D**,**F**,**H**) are provided (**b**, **bb**, **d**, **dd**, **f**, **ff**, **h**, **hh**). Arrows and asterisks indicate the thickness of the cartilage in the femoral condyle (**b**, **d**, **f**, **h**). Asterisks indicate cellularity in the synovium (bb, dd, ff, hh). (**I**) OARSI scores (* *p*-value < 0.05).

**Figure 4 ijms-21-06424-f004:**
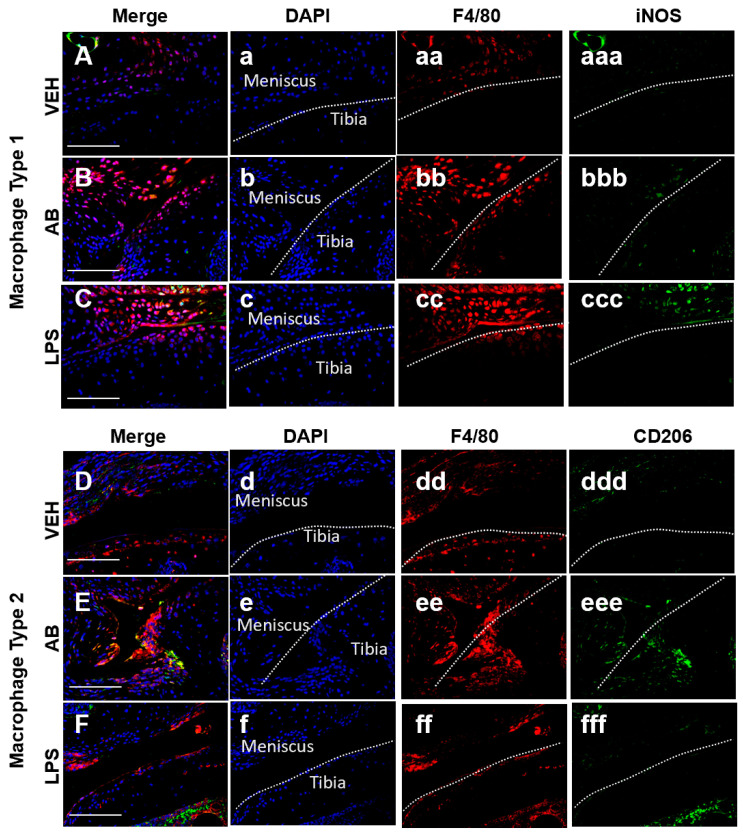
Macrophage infiltration analysis of the VEH-, AB-, and LPS-treated injured joints. (**A**–**C**) Fluorescent immunohistochemistry (IHC) of macrophage type 1 markers F4/80 and iNOS. (**D**–**F**) Fluorescent IHC of macrophage type 2 markers F4/80 and CD205. Dashed line in white represents the surface of the anterior tibial condyle for all images. (**a**–**f**) DAPI; (**aa**–**ff**) F4/80; (**aaa**–**ccc**) iNOS; (**ddd**–**fff**) CD206. White scale bar represents 100 µm.

**Figure 5 ijms-21-06424-f005:**
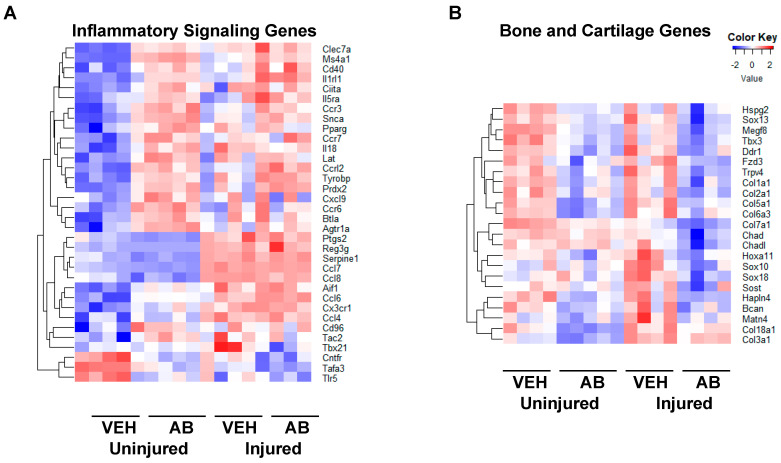
Gene expression changes associated with antibiotic treatment. (**A**) Genes associated with inflammatory process modified by AB treatment and AB with injury compared to VEH uninjured and injured. (**B**) Genes associated with bone and cartilage formation modified by AB treatment along with injury compared to VEH uninjured and injured.

**Table 1 ijms-21-06424-t001:** Inflammatory gene overlap between the AB and VEH groups, showing the expression levels. Fold change (log2 scale) values are shown in the table (FDR < 0.05 for all genes). Changes that were not significant are denoted as ns.

Gene	Uninjured	Injured
***Agtr1a***	0.898	ns
***Aif1***	0.740	ns
***Btla***	0.880	ns
***Ccl4***	1.437	ns
***Ccl6***	0.679	ns
***Ccl7***	−1.600	ns
***Ccl8***	−0.790	ns
***Ccr3***	0.998	ns
***Ccr6***	1.522	ns
***Ccr7***	0.931	ns
***Ccrl2***	0.691	ns
***Cd40***	0.829	ns
***Cd96***	1.028	ns
***Ciita***	1.033	ns
***Clec7a***	0.863	ns
***Cntfr***	−1.117	−0.870
***Cx3cr1***	0.856	ns
***Cxcl9***	0.653	ns
***Il18***	0.846	ns
***Il1rl1***	0.685	0.696
***Il5ra***	0.762	0.719
***Lat***	0.768	ns
***Lat***	0.768	ns
***Ms4a1***	1.293	ns
***Pparg***	0.665	ns
***Prdx2***	0.905	0.607
***Ptgs2***	−0.934	ns
***Reg3g***	−1.769	ns
***Serpine1***	−0.974	ns
***Snca***	1.135	ns
***Tac2***	0.789	ns
***Tafa3***	−1.234	−0.905
***Tbx21***	ns	−0.913
***Tlr5***	−0.987	ns
***Tyrobp***	0.645	ns

**Table 2 ijms-21-06424-t002:** Bone and cartilage gene overlap between the AB and VEH groups, showing expression levels. Fold change (log2 scale) values are shown in the table (FDR < 0.05 for all genes). Changes that were not significant are denoted as ns.

Gene	Uninjured	Injured
***Bcan***	−0.921	−1.072
***Chad***	ns	−0.910
***Chadl***	ns	−0.617
***Col18a1***	−0.923	ns
***Col1a1***	−0.740	−0.782
***Col2a1***	−0.608	−0.752
***Col3a1***	−0.666	ns
***Col5a1***	−0.800	−0.660
***Col6a3***	−1.055	−0.598
***Col7a1***	−0.689	ns
***Ddr1***	ns	−0.603
***Fzd3***	ns	−0.624
***Hapln4***	−0.903	−0.988
***Hoxa11***	ns	−0.696
***Hspg2***	−0.805	−0.660
***Matn4***	ns	−0.596
***Megf8***	−0.693	−0.615
***Sost***	ns	−0.655
***Sox10***	ns	−1.405
***Sox13***	ns	−0.705
***Sox18***	ns	−0.826
***Tbx3***	−0.846	−0.750
***Trpv4***	ns	−0.627

**Table 3 ijms-21-06424-t003:** Overlap of upregulated genes between the AB and LPS groups compared to the VEH showing the expression levels. Fold change (log2 scale) values are shown in the table (FDR < 0.05 for all genes). Changes that were not significant are denoted as ns.

Gene	VEH Uninjured		VEH Injured	
	AB Uninjured	LPS Uninjured	AB Injured	LPS Injured
***4921531C22Rik***	0.680	0.945	ns	ns
***Aldh3b2***	1.289	0.963	ns	ns
***Aldh3b3***	0.783	1.204	ns	ns
***Alox15***	0.652	0.674	ns	ns
***Arhgef3***	0.592	0.602	ns	ns
***Arl11***	0.652	1.176	ns	ns
***Ccl4***	1.437	2.782	ns	ns
***Ccl6***	0.679	1.222	ns	ns
***Cd209a***	1.336	0.901	−0.592	−0.669
***Cd3d***	1.286	1.012	0.858	ns
***Cd84***	0.673	0.776	ns	ns
***Cd8b1***	1.480	ns	ns	ns
***Clec7a***	0.863	1.204	ns	ns
***Csta2***	1.172	1.269	ns	ns
***Cstdc4***	1.338	1.411	ns	ns
***Cx3cr1***	0.856	1.127	ns	ns
***Cyp2ab1***	1.010	1.803	ns	ns
***Fgfbp1***	ns	ns	1.518	1.753
***Gzma***	1.587	1.392	0.610	ns
***Jaml***	0.638	1.826	ns	ns
***Klrc2***	0.784	1.324	ns	ns
***Lep***	−0.701	−0.682	ns	ns
***N4bp2l1***	0.685	0.780	ns	ns
***Nat8l***	0.729	1.402	ns	ns
***Neurl3***	0.782	0.984	ns	ns
***P2rx2***	1.304	1.411	ns	ns
***Ptpro***	0.780	1.169	ns	ns
***Rab20***	ns	ns	0.702	0.608
***Rspo1***	0.768	1.592	−0.751	ns
***Sirpb1a***	0.803	1.056	ns	ns
***Sirpb1b***	0.667	1.096	ns	ns
***Sirpb1c***	0.780	1.109	ns	ns
***Skint3***	0.922	1.670	ns	ns
***Tmem71***	0.745	1.000	ns	ns
***Tnnc1***	ns	ns	0.881	0.780
***Tyrobp***	0.645	0.695	ns	ns
***Vnn3***	0.941	1.210	ns	ns

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
