# Peer review of "Antibiotic Treatment Prior to Injury Improves Post-Traumatic Osteoarthritis Outcomes in Mice"

_ijms, 2020, doi:10.3390/ijms21176424_

Round 1

Reviewer 1 Report

The authors describe the effect of antibiotic therapy on PTOA. Overall, the manuscript is well written. Few suggestions:

Page 2, line 48-“Studies have suggested…”: pls add appropriate references.

Figure 3 isn’t discussed in the text. Moreover, Figure 3 is a part of Figure 5-pls omit out Figure 3 and update the figure numbering.

Section 2.3-pls elaborate on macrophage marker F4/80 in the text.

There are several genes listed in Table 1 that are downregulated. Pls comment on such genes in the Discussion section.

Concluding line on Tlr5 being identified as a therapeutic target: This claim is partially based on previous studies on LPS. I would suggest adding the results where authors would be examining the effect of TLR5 antagonists. Without these experiments/results, it is difficult to confirm the role of TLR5.

Author Response

Page 2, line 48-“Studies have suggested…”: pls add appropriate references.

References have been added.

Figure 3 isn’t discussed in the text. Moreover, Figure 3 is a part of Figure 5-pls omit out Figure 3 and update the figure numbering.

Modification to the figure and text have been done in order to discuss the figure in the text.

        Section 2.3-pls elaborate on macrophage marker F4/80 in the text.

Lines 165-169 the following two sentences have been added ‘Macrophages can be found using the marker F4/80, which is a marker for cells of mononuclear phagocyte lineage in mice [43].’ ‘Identification of both macrophage populations was done using F4/80 and iNOS as an M1 marker while using CD206 as a M2 marker.’

        There are several genes listed in Table 1 that are downregulated. Pls comment on such genes in the Discussion section.

The following language has been added in discussion ‘Gene expression data indicates that inflammatory genes Ptgs2, Reg3g, and Serpine1 were down-regulated in uninjured AB treated while Tbx21 was found to be down-regulated in injured AB joints. Two inflammatory genes found to be down-regulated in both injured and uninjured AB compared to the corresponding VEH were Tafa3 and Cntfr, which ‘are both associated with the immune and nervous system. These genes have not been studied in the context of PTOA and would make interesting candidates for studying their influence on injury outcome.’

        Concluding line on Tlr5 being identified as a therapeutic target: This claim is partially based on previous studies on LPS. I would suggest adding the results where authors would be examining the effect of TLR5 antagonists. Without these experiments/results, it is difficult to confirm the role of TLR5.

The following language has been added in discussion ‘Kim et al showed that tlr5 in rheumatoid arthritis promotes monocyte presence and osteoclast formation due to the cross regulation of the tlr5 and TNF-a pathways [73].’

Reviewer 2 Report

While detailed histology analysis was performed there is no information on the gut microbiome changes, qualitative and quantitative. The authors should provide information/discussion on gut microbiome species that are eliminated after the antibiotic treatment they used. In my opinion, without this information the conclusions are too vague and are hard to translate to humans.

Author Response

        While detailed histology analysis was performed there is no information on the gut microbiome changes, qualitative and quantitative. The authors should provide information/discussion on gut microbiome species that are eliminated after the antibiotic treatment they used. In my opinion, without this information the conclusions are too vague and are hard to translate to humans.

Our study employed the exact antibiotic treatment regimen as several previously published reports that also described the antibiotic cocktail's effects on the gut microbiome, and the impact on bone. For clarification we have added references to these manuscripts in the discussion and they include Rios-Arce et al 2020 (PMID: 32061677) and Guss et al 2017 (PMID: 28244143).

Round 2

Reviewer 1 Report

Only Figure 3dd is discussed in the main text while in section 2.3, Figure 3 is mistakenly numbered as Figure 4. Please correct the figure numbers. 

The concluding paragraph mention that "This study identified a potential therapeutic target, Tlr5...", however in the text/response, the authors clarified that this was identified by Kim et al. Since the role of Tlr5 antagonists will be studied in future, this claim is not supported by the current manuscript/studies. If the authors are not willing to perform the suggested experiments, then please edit the conclusions accordingly. 

Author Response

Only Figure 3dd is discussed in the main text while in section 2.3, Figure 3 is mistakenly numbered as Figure 4. Please correct the figure numbers. 

corrects made

The concluding paragraph mention that "This study identified a potential therapeutic target, Tlr5...", however in the text/response, the authors clarified that this was identified by Kim et al. Since the role of Tlr5 antagonists will be studied in future, this claim is not supported by the current manuscript/studies. If the authors are not willing to perform the suggested experiments, then please edit the conclusions accordingly. 

Sentence has been edited to read: ' This study highlights the importance of the gut biome in modulating PTOA, specifically affecting toll-like receptors and with that changing the PTOA outcome after injury.'

Reviewer 2 Report

None

Author Response

no comments to address